# Imidacloprid Impairs Glutamatergic Synaptic Plasticity and Desensitizes Mechanosensitive, Nociceptive, and Photogenic Response of *Drosophila melanogaster* by Mediating Oxidative Stress, Which Could Be Rescued by Osthole

**DOI:** 10.3390/ijms231710181

**Published:** 2022-09-05

**Authors:** Chuan-Hsiu Liu, Mei-Ying Chen, Jack Cheng, Tsai-Ni Chuang, Hsin-Ping Liu, Wei-Yong Lin

**Affiliations:** 1Graduate Institute of Chinese Medicine, China Medical University, Taichung 40402, Taiwan; 2School of Chinese Medicine, China Medical University, Taichung 40402, Taiwan; 3Graduate Institute of Integrated Medicine, China Medical University, Taichung 40402, Taiwan; 4Department of Medical Research, China Medical University Hospital, Taichung 40402, Taiwan; 5Graduate Institute of Acupuncture Science, China Medical University, Taichung 40402, Taiwan

**Keywords:** Imidacloprid, glutamate receptor, neuroplasticity, *Drosophila melanogaster*

## Abstract

Background: Imidacloprid (IMD) is a widely used neonicotinoid-targeting insect nicotine acetylcholine receptors (nAChRs). However, off-target effects raise environmental concerns, including the IMD’s impairment of the memory of honeybees and rodents. Although the down-regulation of inotropic glutamate receptor (iGluR) was proposed as the cause, whether IMD directly manipulates the activation or inhibition of iGluR is unknown. Using electrophysiological recording on fruit fly neuromuscular junction (NMJ), we found that IMD of 0.125 and 12.5 mg/L did not activate glutamate receptors nor inhibit the glutamate-triggered depolarization of the glutamatergic synapse. However, chronic IMD treatment attenuated short-term facilitation (STF) of NMJ by more than 20%. Moreover, by behavioral assays, we found that IMD desensitized the fruit flies’ response to mechanosensitive, nociceptive, and photogenic stimuli. Finally, the treatment of the antioxidant osthole rescued the chronic IMD-induced phenotypes. We clarified that IMD is neither agonist nor antagonist of glutamate receptors, but chronic treatment with environmental-relevant concentrations impairs glutamatergic plasticity of the NMJ of fruit flies and interferes with the sensory response by mediating oxidative stress.

## 1. Introduction

Imidacloprid (IMD), developed in 1991, is a widely used neonicotinoid to control crop pests and increase crop yield [1]. Neonicotinoids, literally “new nicotine-like insecticides”, account for more than 25% of total pesticides [2], with annual global sales exceeding USD 3.5 billion [3], and IMD alone occupies a 41.5% share of neonicotinoid insecticides. IMD’s action on insect nicotine acetylcholine receptors (nAChRs) is complex. On the one hand, as a partial agonist, IMD triggers sustained electrical impulses of nAChRs [4]. On the other hand, IMD competes with acetylcholine for the nAChRs binding sites and behaves like an antagonist, thus blocking normal acetylcholine signal transduction [4]. Either way, IMD interferes with nAChR transduction and eventually leads to organismal death.

With the widespread use of IMD, there is growing evidence that IMD residues increase the potential risk to non-target organisms, including humans, through food chain transmission [5]. Out of the agriculturally applied IMD, only about 5% are absorbed by crops [6], while most are dispersed into the environment, causing pollution and continuous accumulation in soil [7]. Due to its hydrophilic property, after being applied to the crop, IMD distributes thoroughly across plant tissues [8], which cannot be effectively removed by washing peels or surfaces [9]. It was found that a variety of new nicotine pesticides are commonly detectable in vegetables, fruits, and honey [10], among which the residue of IMD in green pepper is 7.2 ppb [9]. Moreover, IMD concentration in the environment ranges from 0.001 to 320 ppb [11], with the highest residues in surface water [12]. The estimated maximum human chronic daily intake (CDI) through drinking water is 10.2 ng/kg/day [13]. As a result, IMD is ubiquitous from water sources to kitchen taps, and human beings may ingest the pesticide by consuming contaminated food and water [14], which shows a high risk of daily exposure to IMD. Most exposure studies of IMD set the concentrations from 2.5 to 90 ppm [15,16]. However, relatively few studies selected their dose based on the estimated dietary intake of the IMD level of human beings, i.e., 7.2–300 ppb [9,14].

Neonicotinoids effectively control agricultural pests and, unfortunately, also affect off-target organisms, such as bees, invertebrates, and even human beings [10]. IMD generates reactive oxygen species (ROS), causing oxidative stress, interfering with immunity, courtship behavior, and inducing neurological and metabolic disorders in the *Drosophila* animal model [15,17]. Other potential hazards include disrupting reproductive and endocrine systems of mice [16,18], altering body weight over multiple generations of rats [19], causing birth defects, and inducing mutagenesis [16]. In addition, despite its original insect-targeting design, IMD has been shown to partially activate human α4β2 nAChR and antagonize the neurotransmitter acetylcholine (Ach) using electrophysiological recording on HEK cells [20].

Ligand-gated ion channels control the influx or efflux of specific ions across the cell membrane upon the binding of specific ligands and result in polarization or depolarization of the cell. This phenomenon is critical for neuron signal transmission and functional regulation [21]. The main target of IMD, i.e., nAChRs, belongs to a superfamily of cys-loop ligand-gated ion channels, including γ-aminobutyric acid (GABA), glycine, selective serotonin receptor (5-HT3), and zinc-activated receptors [22]. In addition to nAChRs, IMD also affects GABA receptors by blocking GABA; hence, IMD may act as an antagonist of GABA receptors [4]. GABA receptors control the neuronal inhibition in the central nervous system of vertebrates and are associated with neurological diseases, including autism, epilepsy, schizophrenia, and addiction [23]. Interestingly, the association between IMD exposure and autism has been reported [24]. Moreover, IMD exposure impairs the memory of both invertebrates and vertebrates, including honeybees [25], bats [26] and rats [27]. The down-regulation of inotropic glutamate receptor (iGluR) GRIN1 in the rat hippocampus was identified along with the memory impairment [27]. It is well established that inotropic glutamate receptors contribute to synaptic transmission, neural plasticity, and memory [28]. However, whether IMD directly manipulates the activation or inhibition of glutamate receptors is unknown.

The neuromuscular junction (NMJ) of the fruit fly (*Drosophila melanogaster*) is a perfect model to investigate the IMD–iGluR interaction. In contrast to the acetylcholinergic signaling used by the vertebrate NMJ, the fruit fly NMJ uses iGluR for transduction [29]. Furthermore, the electrophysiological recording for fruit fly NMJ and its glutamatergic properties are well established [30]. Moreover, the results can be transferred to the honeybee, which is the major focus of the research concerning the environmental residue of IMD. In addition to verifying the IMD–iGluR interaction, we also investigated the NMJ plasticity and other behavioral traits related to the peripheral nervous system, including mechanosensation, nociception, and phototransduction. Finally, we tested whether IMD-exerting oxidative stress mediates the behavioral phenotypes by co-treatment with the antioxidant osthole (OST, 7-methoxy-8-(3-methylbut-2-enyl)chromen-2-one).

## 2. Results

### 2.1. Determine the IMD Dose

To determine the optimal IMD dose for chronic treatment for the fruit fly, we used the survival assay to screen the harmless dose. As shown in Figure 1A, the survival rate due to the dose of 0.125 mg/L is indistinguishable from that of the control. We further confirmed continuous exposure to a dose of 0.125 mg/L from the embryo stage resulted in a non-significant change to the body weight of both male and female fruit flies (Figure 1B), which may imply this dose harbors little or no harm to the metabolism and development of the fruit fly. Therefore, the dose of 0.125 mg/L was used in the following chronic treatment experiments. Notably, 0.125 mg/L falls in the range of the estimated environmental IMD residue and the estimated dietary intake of IMD level of human beings.

### 2.2. IMD Is Neither Agonist Nor Antagonist of Glutamate Receptor

As mentioned earlier, despite the sufficiency of comprehensive studies of IMD on nAChR and GABAR, there is a lack of knowledge of IMD on glutamate-mediated neural transmission, especially the interaction of IMD with the glutamate receptor. We chose the neuromuscular junction (NMJ) of the fruit fly larva as the experiment model [29], since in contrast to the acetylcholine-mediated vertebrate motor neurons, the counterpart of the fruit fly uses glutamate as the neurotransmitter. With the conventional intracellular recordings by bridge mode [30], the voltage change (∆V_M_) of the membrane potential driven by an air pressure ejection (puff) of chemical on the NMJ was recorded. As shown in Figure 1C, as expected, glutamate triggered a dose-dependent ∆VM on the NMJ. However, IMD of the dose of 0.125 mg/L and even two orders of magnitude higher could not trigger any slight ∆V_M_ (Figure 1D). Moreover, after the IMD puffs, an additional glutamate puff was given, and the glutamate-triggered ∆VM was the same as the pre-IMD glutamate puffs (Figure 1E). To confirm whether IMD antagonizes glutamate, NMJ preparations were stimulated by glutamate puffs under the presence of IMD (Figure 1F). We found that IMD did not antagonize glutamate (Figure 1G). An agonist is a chemical that excites a receptor, while an antagonist blocks it. Therefore, IMD is neither agonist nor antagonist of the glutamate receptor of the fruit fly.

### 2.3. Chronic IMD Treatment Damps the Glutamatergic Synaptic Plasticity

Although IMD is neither agonist nor antagonist of the glutamate receptor, one could not rule out the possibility that chronic IMD treatment affects glutamatergic neurotransmission. This hypothesis was tested by recording the short-term facilitation (STF) of the glutamatergic NMJ [31] using the two-electrode voltage-clamp technique [32]. STF is a synaptic phenomenon that enhances synaptic transmission due to previous stimulations and therefore is a form of synaptic plasticity. As shown in Figure 2A, the consecutive 25 Hz pulse-evoked junction current of NMJ was recorded, and chronic IMD treatment damped the STF (Figure 2B). In another program setting, a series of pulses with the frequency gradually increasing from 0.5 to 20 Hz was applied (Figure 2C) and again confirmed the damped STF by chronic IMD treatment (Figure 2D). Therefore, while IMD is neither agonist nor antagonist of the glutamate receptor, the chronic IMD treatment damped the glutamatergic synaptic plasticity.

### 2.4. Chronic IMD Treatment Evokes Nerve-Conduction-Associated Behavioral Phenotypes

Variant ionotropic glutamate receptors [33] along with TRP channels [34] complementarily mediate peripheral mechanosensation, thermosensation, and photosensation in the fruit fly. Therefore, we further explored whether chronic IMD treatment interferes with these nerve-conduction-associated behaviors.

Upon gentle touch on the mouthparts, the fruit fly larvae show characteristic responsive movement as illustrated in (Figure 3A). Each movement until the recovery of forward-moving is assigned a score, and the sum of the scores represents the degree of mechanosensory response [35]. As shown in Figure 3B, chronic IMD treatment decreased the mechanosensory response. Further RT-qPCR (Figure 3C) showed that the gene expression of Arr2, which mediates both desensitization and resensitization processes of the G-protein coupled receptor (GPCR) [36], was significantly decreased.

Nociception is the perception of painful stimuli and serves as a signal to trigger an appropriate defense response. Avoidance of heat is a classic assay to probe the nociceptive response of flies (Figure 3D). We found that chronic IMD treatment decreased the nociceptive response (Figure 3E) and changed the gene expression TRPM (Figure 3F).

To probe photosensation, electroretinogram (ERG) was used to record the electrophysiological signal of the fruit fly compound eye upon on/off light stimuli. From the wave form of the ERG signal, many physiological parameters could be deduced (Figure 3G). We found that total retinal voltage change (∆V_R_) and the receptor potential amplitude (RPA), which represents the light-induced depolarization of the photoreceptor neuron, were decreased upon chronic IMD treatment (Figure 3H). This result reproduced the ERG phenotype induced by IMD at a much higher dose [15]. We further showed that gene expression in the phototransduction pathway was vastly altered (Figure 3I), including Guanine nucleotide-binding proteins (G proteins), eye-specific protein kinase C (PKC) proteins (e.g., inaC, inaD, and inaE), and a transient receptor potential channel trpl.

### 2.5. IMD-Induced Phenotypes Were Rescued by Antioxidant Osthole

It is well established that IMD harms the exposed organism by mediating oxidative stress [37,38], and we confirmed that chronic IMD treatment decreased the expression of oxidative stress responding genes (Figure 4A). Therefore, we tested whether co-treatment with the antioxidant osthole [39] ameliorates STF attenuation and rescues other behavioral phenotypes caused by chronic IMD treatment. We found that osthole rescued IMD-induced alteration of STF (Figure 4B), retinal light response (Figure 4C,D), mechanosensory response (Figure 4E), and nociceptive response (Figure 4F). These results may indicate that IMD interferes with synaptic plasticity and nerve-conduction-associated behaviors at least by mediating oxidative stress.

## 3. Discussion

In this study, we investigated the IMD–iGluR interaction using the fruit fly NMJ model and identified that IMD is neither agonist nor antagonist of iGluR. However, we also found that chronic exposure to the environmental-relevant dose of IMD decreased neuroplasticity of NMJ and interfered with other behavioral traits related to the peripheral nervous system, including mechanosensation, nociception, and phototransduction. We also validated that IMD-exerting oxidative stress mediates these behavioral phenotypes.

Our findings revealed that even though IMD is neither agonist nor antagonist of iGluR, chronic exposure to the environmental-relevant dose of IMD damps the short-term facilitation, a kind of plasticity of the glutamatergic synapse. When an organism is exposed to a continuous stimulus, neural plasticity either enhances or desensitizes the signal according to its inherited genetic program. For insects like honeybees, glutamatergic plasticity participates in food procuring behavior [40] and memory [41]. For vertebrates like human beings, glutamatergic plasticity is involved in reward, cognition [42], addiction [43], learning, and memory [44]. Thus chronic exposure to the environmental IMD may provoke relevant biological issues by mediating synaptic plasticity.

Synaptic transmission underlies stimulus perception, regardless of the types of stimuli. Nociception is the sensation of noxious stimuli and is critical for an organism to avoid a dangerous environment. We showed that IMD treatment reduces thermal nociception and decreases the expression of the transient receptor potential cation channel subfamily M (TRPM). The TRP channel family is conserved from invertebrates to humans and responds to diverse environmental stimuli [45]. TRPM is a temperature-sensitive channel and involves thermal nociception [46]. Thus, besides neuroplasticity, alteration of the expression of temperature-sensitive channels may underlie IMD-induced reduction of nociception.

Mechanosensation is the sensation of mechanical stimuli and is the basis for sensing gentle touch, hearing, and proprioception [47]. We showed that chronic IMD treatment reduces mechanosensory response and decreases the expression of arrestin 2 (Arr2). Arr2 mediates both desensitization and resensitization of G-protein coupled receptor (GPCR) signaling and involves sensory transduction [48]. The decreased Arr2 may slow down the relay of the signal. Thus, in addition to neuroplasticity, alteration of the expression of the GPCR regulator may underlie IMD-induced reduction of mechanosensation. Furthermore, the orthologous gene of Arr2 in Homo sapiens is ARRB2, which encodes the beta-arrestin 2 protein. Importantly, besides mechanosensation, beta-arrestin 2 participates in critical physiological processes, including the regulation of pancreatic, cardiac, and metabolic functions [49]. Previous animal studies showed that deficiency of beta-arrestin 2 contributes to insulin resistance [50] and decreases oxidative stress in hepatic fibrosis [51]. Interestingly, animal studies on IMD revealed its disruption of glucose homeostasis [52] and liver health [53]. Thus, the disruption of Arr2 expression by IMD may underlie IMI’s pathological effects on vertebrates.

Retinal phototransduction is the first step of visual signaling. Vision is vital for diurnal animals, including honeybees and human beings. We showed that chronic IMD treatment reduces the light-induced depolarization of the photoreceptor neuron of the retina, which confirms the findings by [15]. We further found the up-regulation of several genes in the cascade of phototransduction. This may be due to homeostatic gene expression to compensate the IMD-perturbed phototransduction pathway.

Previous studies have shown that IMD induces oxidative stress responses, such as catalase (CAT) and glutathione S-transferase (GST) activity [54] and induces three isoforms of nitric oxide synthases (iNOS, eNOS, nNOS) [55]. These oxidative stresses may cause damage to tissues and neurons, and we found that antioxidant osthole could rescue IMD-induced phenotypes, thus confirming that chronic IMD treatment with environmental-relevant concentration impairs neuroplasticity and peripheral sensations by mediating oxidative stress.

## 4. Materials and Methods

### 4.1. Chemicals and Fly Strain

Imidacloprid (IMD) was purchased from Toronto Research Chemicals (Toronto, ON, Canada) and stored at −20 °C. The IMD-containing medium was prepared by diluting the IMD aliquot into 10 mM stock with Dimethyl sulfoxide (DMSO), purchased from J.T.Baker (Radnor, PA, USA), and added to *Drosophila* medium when the medium had cooled to 55 °C from boiling to final IMD concentrations as wished. The medium was mixed thoroughly with a food processor for one minute before solidification. Extra DMSO was added so that the medium of each IMD concentration, including the control medium, contains the same amount of DMSO. Osthole was kindly provided by Dr. Yueh-Hsiung Kuo of China Medical University, Taiwan. L-glutamate acid was purchased from Sigma-Aldrich (Burlington, MA, USA); 1 M glutamate was prepared by dissolving into distilled water and adjusting to pH 8 with 5N NaOH. To evaluate the efficacy of the rescue treatment, each assay was repeated with three groups of flies, i.e., the control group, the IMD group treated with 0.125 mg/L IMD, and the rescue group co-treated with 0.125 mg/L IMD plus 6 μg/mL osthole. A white (w) stock outcrossed with Canton S 10 times (w (CS10)) was utilized in all experiments. The formula for *Drosophila* medium and the culture environment follows our previous work [56].

### 4.2. NMJ Electrophysiology

#### 4.2.1. Larvae Sample Preparation

Embryos were cultured in the control or 0.125 mg/L IMD-containing medium until 3rd instar larvae. The dissection of larvae for the preparation of body-wall muscles with intact peripheral neural fibers and the mounting of samples for recording followed our previous work [57].

#### 4.2.2. Bridge Mode Recording of Membrane Potential

For membrane potential measurement, the dissected 3rd instar larvae were relocated to HL-3 buffer containing an extra 0.6 mM Ca^2+^, and the NMJ preparation was recorded using bridge mode (AxoClamp 2B, AXON Instruments, Foster City, CA, USA) [30], with the reference electrode containing 3 M potassium chloride and the recording electrode containing 2 M potassium chloride and 0.33 M potassium citrate tribasic. After both electrodes were submerged into the HL-3 buffer, the potential difference between electrodes was reset to zero, and then the recording electrode was gently stabbed into the M12 muscle of the A4 section. A glass capillary for the pneumatic drug ejection system (PDES-02TX, NPI electronic GmbH, Tamm, Germany) was pre-pulled by the micropipette puller (P2000, Sutter Instrument, Novato, CA, USA). Each chemical puff was given pneumatically with 15 psi pressure, 0.3-sec duration, and about 60 sec delay between puffs. For recording to verify IMD as an antagonist, IMD was additionally added to the mounting HL-3 buffer to the concentration as mentioned.

#### 4.2.3. Two-Electrode Voltage-Clamp Recording of STF

For short-term facilitation (STF) measurement, the dissected 3rd instar larvae were relocated to HL-3 buffer containing an extra 0.4 mM Mg^2+^ and 0.2 mM Ca^2+^, and the NMJ preparation was recorded using a two-electrode voltage clamp (AxoClamp 2B, AXON Instruments, Foster City, CA, USA). The setup of the two-electrode voltage clamp followed our previous work [57]. For measuring STF of frequency-increasing stimuli, after confirmation of the stability at the 5th min of recording, consecutive tetanus stimuli were pre-programmed in Clampex software 10.6 (Molecular Devices, San Jose, CA, USA) and given by S88 Stimulator (Grass Instruments, Quincy, Norfolk County, MA, USA) with program triggering. We measured STF upon two stimulation programs. The first one contained tetanus of 20 stimuli of 2 Hz, 5 Hz, 10 Hz, and 20 Hz. Then, 5 consecutive 25 Hz stimuli were delivered, respectively. The EJCs were analyzed by pCLAMP software (Molecular Devices, San Jose, CA, USA). The amplitude of the last 10 EJC of each stimuli frequency were normalized to that of 0.5 Hz stimuli for comparison. For measuring STF of 25 Hz stimuli, tetanus of 5 consecutive stimuli with 40 ms delay, i.e., 25 Hz stimulus, were delivered, and the amplitude of EJCs was normalized to the first EJC of the sample for further comparison. At least eight replicates for each group were observed.

### 4.3. Behavioral Assays

#### 4.3.1. Survival Assay

After three days from emergence, male flies were transferred to vials with culture media containing different concentrations of IMD, with 35 flies in each vial and five vials for each IMD concentration. The number of dead flies was counted every day for seven days.

#### 4.3.2. Body Weight

After three days from emergence, virgin female flies were allowed to mate with male flies for three days in the environment supplied with the control culture medium. Mated flies were transferred to vials with culture media containing different concentrations of IMD as well as tomato juice, with 10 female and 10 male flies in each vial and 5 vials for each IMD concentration. After one day, the mated flies were discarded. After five days from emergence, flies were weighted after CO_2_ anaesthetization using analysis balance (TB-214, Denver Instrument, Arvada, CO, USA).

#### 4.3.3. Electroretinogram (ERG)

Embryos were cultured in the control or 0.125 mg/L IMD-containing medium until the 20th day after emergence. The experiment setup followed our previous study [58]. The ERG waveform of each fruit fly was recorded for six cycles and was analyzed by Axoscope software (Molecular Devices, San Jose, CA, USA). The characteristics of the ERG waveform, including total retinal voltage change (∆V_R_), receptor potential amplitude (RPA), on-transient, and off-transient, are illustrated in Figure 3G. At least 13 replicates for each group were observed.

#### 4.3.4. Mechanosensory Response

The gentle touch response of *Drosophila* larvae was measured as described [35]. Briefly, three-instar larvae were placed on a solidified 2% agar plate after being washed gently with saline solution. Then, the mouth parts of the larvae were touched gently with eye-lash when they were moving forward. The responses until the recovery of forward-moving of the larvae were scored as illustrated in Figure 3A. A total of 40 larvae were assayed for each group.

#### 4.3.5. Nociception

The offspring generation of the IMD-treated flies was used in this assay. Briefly, parental virgin flies were treated with the control or 0.125 mg/L IMD-containing medium for 17 days and were allowed to mate for 3 days. After mating, parent flies were transferred to vials with the control medium for egg laying for another two days. Then, the parent flies were discarded, and the offspring embryos developed to pupae in the environment of the control medium, i.e., the offspring generation was not exposed to IMD directly. After five days of emergence, adult flies of the second generation were subjected to the nociception assay. Briefly, for one replicate, 20 flies were aspirated into a 35 mm plastic dish (Corning, Corning City, NY, USA) from a drilled hole on the cover, and the hole was sealed with transparent tape. The dish was then placed on a pre-heated metal block for one minute in a dark room, where white light lamps were turned off, and observations were carried out under red light. The number of flies left on the bottom and those who relocated themselves to the cover to avoid the heat stimulus was counted. The nociceptive responding rate was defined as the number of flies on the top cover divided by the number of loaded flies; 10 replicates were observed for each group. Notably, for optimal response, female and male flies were tested separately with different temperatures of heat block, 47.3 °C for female while 44.6 °C for male flies.

### 4.4. RNA Extraction and RT-qPCR

For one replicate of RNA extraction, 30 3rd instar larvae, the whole body of 50 male adult flies, or 100 heads of adult flies were collected. Three replicates were collected, and the IMD treatment on larvae, adult flies for the whole body, or adult flies for head followed the recipe used in mechanosensory response, nociception, or ERG assays, respectively. RNA was extracted using PureZOL reagent (Bio-Rad, Hercules, CA, USA) following the manufacturer’s manual. NanoDrop Spectrophotometer (Thermo Fisher Scientific, Waltham, MA, USA) was used to quantify and assess the purity of RNA. RNA was reverse-transcribed using the high-capacity cDNA reverse transcription kit (Applied Biosystems, Waltham, MA, USA). Quantitative PCR reaction was prepared using LightCycler 480 SYBR Green I Master mix (Roche, Basel, Switzerland) and assayed with ViiA 7 Real-Time PCR System (Thermo Fisher Scientific, Waltham, MA, USA). Three replicates for each group were observed. The primers are listed in Appendix A.

### 4.5. Statistics

The Student’s *t*-test was used to estimate the significance of the difference between the control and IMD-treated groups of all assays except survival and STF of NMJ. The log-rank test was used to estimate the significance of the difference between survival curves. The two-way ANOVA with Tukey’s multiple comparison test was used to estimate the significance of the difference between groups of STF of NMJ. One-way ANOVA with Tukey’s multiple comparison test was used to estimate the significance of the difference in rescue assays.

## 5. Conclusions

We clarified that IMD is neither agonist nor antagonist of glutamate receptors, but chronic treatment with an environmental-relevant concentration impairs glutamatergic plasticity of the NMJ of *Drosophila*. Furthermore, IMD interferes with the nervous system, including mechanosensation, nociception, and phototransduction. Moreover, the antioxidant osthole rescued the IMD-induced phenotypes and confirmed that chronic IMD treatment provokes these phenotypes by mediating oxidative stress.

## Figures and Tables

**Figure 1 ijms-23-10181-f001:**
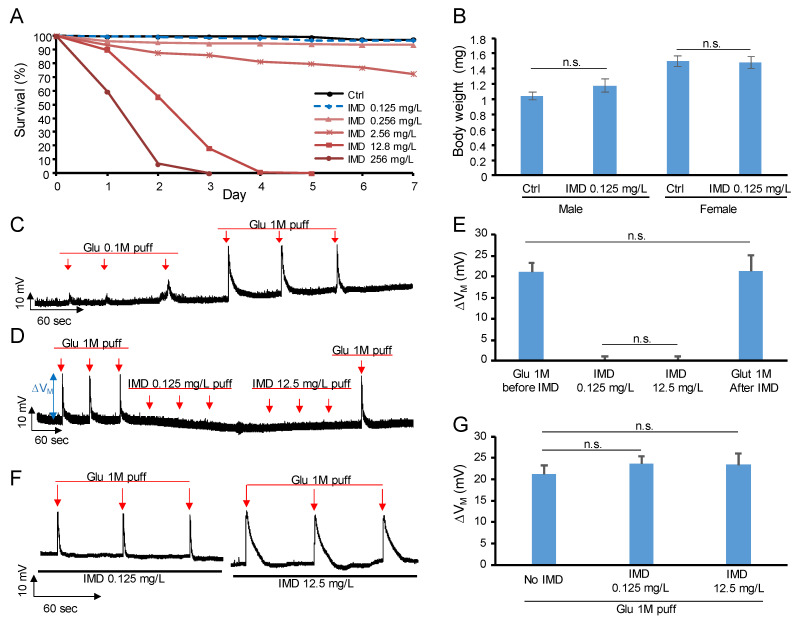
The selection of the dose of IMD and the acute application on the glutamatergic neuromuscular junction (NMJ) of the fruit fly: (**A**) survival of fruit flies receiving different doses of IMD; (**B**) the dose of 0.125 mg/L did not affect the body weight of the flies; (**C**) the representative voltage recording showing the puff of glutamate on NMJ triggered a dose-dependent synaptic voltage change; (**D**) the representative voltage recording showing two doses of IMD puff did not trigger voltage change of NMJ nor decrease the amplitude of the following glutamate puff; (**E**) the quantification of (**D**); (**F**) the representative voltage recording showing two doses of IMD did not antagonize glutamate puff; (**G**) the quantification of (**F**). The error bar stands for the standard error of the mean. The red arrows stand for the puff of chemicals on NMJ. ∆V_M_, voltage change of the membrane potential; IMD, imidacloprid; Glu, glutamate. n.s., not significant.

**Figure 2 ijms-23-10181-f002:**
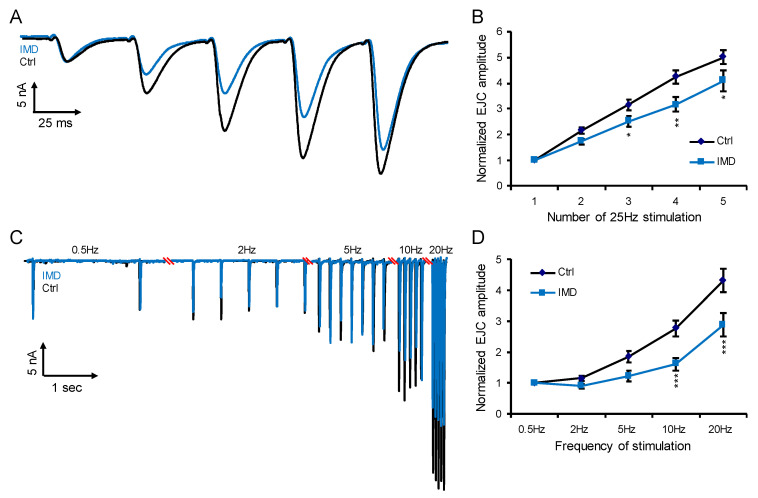
Chronic IMD treatment attenuated the short-term facilitation (STF) of the glutamatergic neuromuscular junction (NMJ) of the fruit fly: (**A**) the representative recording showing the evoked currents by five consecutive electric pulses of the frequency of 25 Hz; (**B**) the quantification of (**A**); (**C**) the representative recording showing the evoked currents by electric pulse with increasing frequency from 0.5 to 20 Hz; (**D**) the quantification of (**C**). The error bar stands for the standard error of the mean. The *, **, or *** denotes the *p*-value < 0.05, 0.01, or 0.001 for Two-way ANOVA with Tukey’s multiple comparison test. IMD, imidacloprid of 0.125 mg/L.

**Figure 3 ijms-23-10181-f003:**
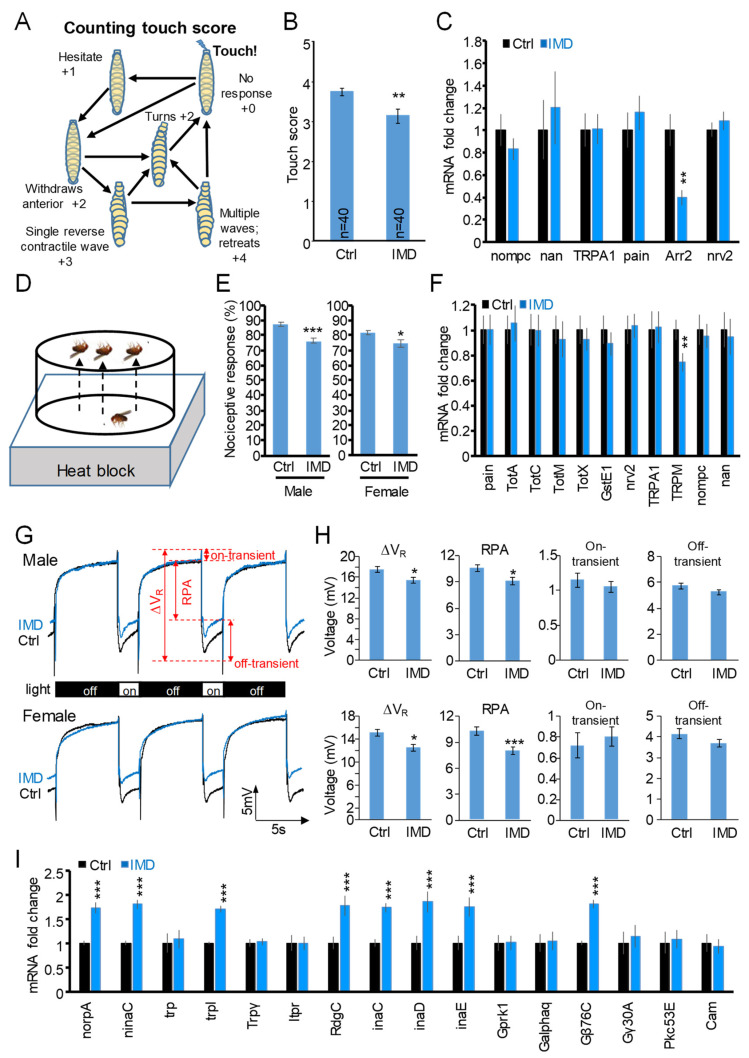
Chronic IMD treatment evoked behavioral phenotypes of the fruit fly. IMD caused alteration of the mechanosensory response: (**A**) scheme showing touch score counting; (**B**) the touch score; (**C**) gene expression of the mechanosensory pathway; IMD decreased the nociceptive response; (**D**) diagram showing the nociception assay; (**E**) the nociception behavior of male and female flies; (**F**) gene expression of the nociception pathway; IMD causes alteration of the light perception; (**G**) the representative waveform of electroretinogram (ERG) upon light on-off stimuli; (**H**) the quantification of ERG waveform; (**I**) gene expression of the phototransduction pathway. Error bar stands for standard error of the mean. The *, **, or *** denotes the *p*-value < 0.05, 0.01, or 0.001 for Student’s *t*-test; IMD, imidacloprid of 0.125 mg/L; ∆V_R_, voltage change of electroretinogram; RPA, receptor potential amplitude. The operational definition of ∆V_R_, RPA, on-transient, and off-transient is illustrated in (**G**).

**Figure 4 ijms-23-10181-f004:**
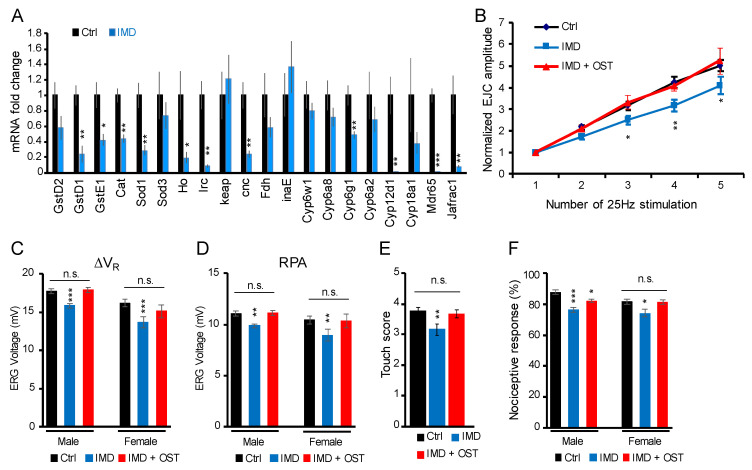
The antioxidant osthole (OST) rescued the phenotypes of the IMD-treated fruit fly: (**A**) differentially expressed genes of the oxidative stress response pathway; (**B**) the rescued 25 Hz short-term facilitation (STF) of the neuromuscular junction; (**C**) the rescued retinal ∆V_R_ and (**D**) RPA; (**E**) the rescued mechanosensory perception; (**F**) the rescued nociceptive perception. Error bar stands for standard error of the mean. The *, **, or *** denotes the *p*-value < 0.05, 0.01, or 0.001 and n.s. for not significant of Student’s *t*-test (**A**); two-way ANOVA with Tukey’s multiple comparison test (**B**); and one-way ANOVA with Tukey’s multiple comparison test (**C**–**F**); IMD, imidacloprid of 0.125 mg/L; OST, osthole of 6 μg/mL; ∆V_R_, voltage change of electroretinogram; RPA, receptor potential amplitude.

## Data Availability

The data presented in this study are openly available in FigShare at https://doi.org/10.6084/m9.figshare.20367312.v1 (accessed on 27 July 2020).

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
