# Peer review of "Imidacloprid Impairs Glutamatergic Synaptic Plasticity and Desensitizes Mechanosensitive, Nociceptive, and Photogenic Response of Drosophila melanogaster by Mediating Oxidative Stress, Which Could Be Rescued by Osthole"

_ijms, 2022, doi:10.3390/ijms231710181_

Round 1

Reviewer 1 Report

The article titled Imidacloprid impairs glutamatergic synaptic plasticity and desensitizes mechanosensitive, nociceptive, and photogenic response of Drosophila melanogaster by mediating oxidative stress, which could be rescued by osthole” submitted by Chuan-Hsiu Liu et al., is interesting.  The authors employed the Drosophila melanogaster as a model to investigate the toxicity of Imidacloprid (IMD), a commonly used neonicotinoid, with a focus on the molecular level. These findings are interesting and the manuscript is well-written and, well-structured.  I believe that the study's objectives are interesting and fit well within the scope of the journal and that the analysis was carried out with an appropriate description of the methods utilized.

Comments to authors

What is (Bass et al., 2015)? (Line 33)

Lines 58-64 would be clearer if the authors mentioned on which animal model the Imidacloprid toxicological assessment was performed. 

Author Response

Q: What is (Bass et al., 2015)? (Line 33)

A: Thanks a lot. This reference is provided as [58] in the revised manuscript.

Q: Lines 58-64 would be clearer if the authors mentioned on which animal model the Imidacloprid toxicological assessment was performed. 

A: Thanks a lot. The experimental models are provided.

Reviewer 2 Report

The current study attempts to assess the effect of Imidacloprid, a systemic insecticide which, along with its high water solubility, creates the potential for the exposure of insects consuming pollen and plant tissues. C-H Liu et al. found that Imidacloprid is neither agonist nor antagonist of glutamate receptors, but chronic treatment with environmental-relevant concentration im pairs glutamatergic plasticity of the NMJ of fruit flies and interferes with the sensory response by mediating oxidative stress.

I found the paper to be overall very well written and I felt confident that the authors performed careful literature analysis and research data interpretation. I recommend that only a minor revision of the manuscript is warranted. I explain my concerns in more detail below and I would ask that the authors will correct them.

Minor comments:

Line 49 : the text in bold is not necessary.

Line 65: please introduce first the name and then the abbreviation; maybe some readers will not know that ACh is the neurotransmitter acetylcholine.

Line 70: again, please introduce the name of 5-HT3 abbreviation – as selective serotonin receptor.

Line 165: “Avoidance of heat is a classical assay to probe the nociceptive response of flies” – please indicate the temperature used for the heat block represented in the figure 3D. Use brackets (temp).

Line 278:  3rd !

Line283 & Line 296: please write Mg 2+ and Ca 2+ using 2+ as superscript !

Line 322: CO2 , write 2 as subscript !

Author Response

Line 49 : the text in bold is not necessary.

A: We cannot see the text in bold in Line 49. As a guess, we deleted "as high as".

Line 65: please introduce first the name and then the abbreviation; maybe some readers will not know that ACh is the neurotransmitter acetylcholine.

A: Thanks a lot. Modified as indicated.

Line 70: again, please introduce the name of 5-HT3 abbreviation – as selective serotonin receptor.

A: Thanks a lot. Modified as indicated.

Line 165: “Avoidance of heat is a classical assay to probe the nociceptive response of flies” – please indicate the temperature used for the heat block represented in the figure 3D. Use brackets (temp).

A: The experimental details of the nociceptive response assay was provided in the Methods section 4.3.5. The temperature used for the heat block: "Notably, for optimal response, female and male flies were tested separately with different temperatures of heat block, 47.3℃ for female while 44.6℃ for male flies."

Line 278:  3rd !

A: Thanks a lot. Modified as indicated.

Line283 & Line 296: please write Mg 2+ and Ca 2+ using 2+ as superscript !

A: Thanks a lot. Modified as indicated.

Line 322: CO2 , write 2 as subscript !

A: Thanks a lot. Modified as indicated.

Round 2

Reviewer 1 Report

The authors modified the draft and it is acceptable. 

Reviewer 2 Report

Dear authors,

Thank you for completed the previously requested corrections.